# Evaluate the Differences in CT Features and Serum IgG4 Levels between Lymphoma and Immunoglobulin G4-Related Disease of the Orbit

**DOI:** 10.3390/jcm9082425

**Published:** 2020-07-29

**Authors:** Wei-Hsin Yuan, Anna Fen-Yau Li, Shu-Yi Yu, Ying-Yuan Chen, Chia-Hung Wu, Hui-Chen Hsu, Jiing-Feng Lirng, Wan-You Guo

**Affiliations:** 1Division of Radiology, Taipei Municipal Gan-Dau Hospital (Managed by Taipei Veterans General Hospital), Taipei 11260, Taiwan; 2Department of Radiology, Taipei Veterans General Hospital, Taipei 11217, Taiwan; frankfbo@gmail.com (S.-Y.Y.); chwu16@vghtpe.gov.tw (C.-H.W.); jflirng@vghtpe.gov.tw (J.-F.L.); wyguo@vghtpe.gov.tw (W.-Y.G.); 3School of Medicine, National Yang-Ming University, Taipei 10556, Taiwan; Fyli@vghtpe.gov.tw (A.F.-Y.L.); yychen354@gmail.com (Y.-Y.C.); 4Department of Pathology, Taipei Veterans General Hospital, Taipei 11217, Taiwan; 5Division of Radiology, National Yang-Ming University Hospital, Yilan City 26058, Taiwan; 6Department of Medical Imaging, Taiwan Adventist Hospital, Taipei 10556, Taiwan; hueichenhsu@gmail.com

**Keywords:** immunoglobulin G4-related orbital disease (IgG4-ROD), orbital lymphoma (OL), computed tomography (CT), Hounsfield unit

## Abstract

Background: Benign immunoglobulin G4 (IgG4)-related orbital disease (IgG4-ROD)—characterized as tumors mimicking malignant orbital lymphoma (OL)—responds well to steroids, instead of chemotherapy, radiotherapy and/or surgery of OL. The objective of this study was to report the differences in computed tomography (CT) features and- serum IgG4 levels of IgG4-ROD and OL. Methods: This study retrieved records for patients with OL and IgG4-ROD from a pathology database during an eight-year-and-five-month period. We assessed the differences between 16 OL patients with 27 lesions and nine IgG4-ROD patients with 20 lesions according to prebiopsy CT features of lesions and prebiopsy serum IgG4 levels and immunoglobulin G (IgG) levels This study also established the receiver-operating curves (ROC) of precontrast and postcontrast CT Hounsfield unit scales (CTHU), serum IgG4 levels, serum IgG levels and their ratios. Results: Significantly related to IgG4-ROD (all *p* < 0.05) were the presence of lesions with regular borders, presence of multiple lesions—involving both lacrimal glands on CT scans—higher median values of postcontrast CTHU, postcontrast CTHU/precontrast CTHU ratios, serum IgG4 levels and serum IgG4/IgG level ratios. Compared to postcontrast CTHU, serum IgG4 levels had a larger area under the ROC curve (0.847 [95% confidence interval (CI): 0.674–1.000, *p* = 0.005] vs. 0.766 [95% CI: 0.615–0.917, *p* = 0.002]), higher sensitivity (0.889 [95% CI: 0.518–0.997] vs. 0.75 [95% CI: 0.509–0.913]), higher specificity (0.813 [95% CI: 0.544–0.960] vs. 0.778 [95% CI: 0.578–0.914]) and a higher cutoff value (≥132.5 mg/dL [milligrams per deciliter] vs. ≥89.5). Conclusions: IgG4-ROD showed distinct CT features and elevated serum IgG4 (≥132.5 mg/dL), which could help distinguish IgG4-ROD from OL.

## 1. Introduction

Orbital space-occupying lesions comprise a wide range of benign and malignant masses [1]. The top eight ocular adnexal lesions include lymphoid tumors, inflammatory disease, cavernous hemangioma, lymphangioma, meningioma, optic nerve glioma, metastatic breast cancer and capillary hemangioma [2]. Several studies indicate that ocular adnexal lymphomas almost account for up to half of all malignant orbital lesions in adults [2,3,4].

Immunoglobulin G4–related disease—which can involve any organs including orbital structures—is a systemic fibroinflammatory condition due to tissue infiltration by immunoglobulin G4 (IgG4) plasma cells [5]. IgG4-related disease in orbit tends to form tumor-like lesions, which are difficult to differentiate from intraorbital lymphoma because both are tumors rich in lymphoplasmacytic infiltration [6]. Lymphoma, one of the most common orbital malignancies in adults [3,4], needs radiotherapy, systemic chemotherapy and/or surgery [7,8]. In contrast, approximately 90% of patients with IgG4-related orbital disease (IgG4-ROD) respond well to steroid therapy [5] Therefore, rapid and accurate diagnosis of IgG4-ROD to help patients receive early steroid treatment is critical.

IgG4-related disease diagnostic criteria commonly follow: a serum IgG4 concentration higher than 135 mg/dL (milligrams per deciliter) [9], the ratio of IgG4-positive/immunoglobulin G (IgG)-positive plasma cell (IgG4+/IgG+ ratio) is >40% or IgG4+ cells > 10/high-powered field of biopsy sample [9]. However, up to 40% of patients with IgG4-related disease may have serum IgG4 levels within the normal range [10]. Pathology and immunohistochemistry remain the gold standard for accurate diagnosis of IgG4-related disease [5,11].

Furthermore, orbital neoplasm rupture via biopsy may lead to tumor seeding and poor prognosis [12]. Integrating clinical findings, serologic data and radiological features is important to establish the prebiopsy diagnosis of IgG4-ROD [12]. Computed tomography (CT) scans provide rapid high-resolution images of orbits for radiological feature extraction [13]. As such, this study integrates CT qualitative and quantitative (Hounsfield unit density) features, serum IgG, and IgG4 levels to differentiate IgG4-ROD from orbital lymphoma (OL).

## 2. Material and Methods

### 2.1. Patients

The Institutional Review Board of Taipei Veterans General Hospital (TVGH) approved this study to waive informed consent because of the retrospective nature of the research.

A doctor (SYY) blinded from the research hypothesis searched pathologic results from the pathology database at TVGH using the keywords “orbit” or “orbital” for cases from 1 January 2010, to 31 May 2018.

The research returned 178 patients with orbital lesions and pathologic results. Thirteen (7%) of 178 patients had multiple orbital lesions (≥2): one (7.7%) of the thirteen patients with lung carcinoid tumors and multiple metastases in the right orbital cavity and 12 (92.3%) patients with lymphoplasmacyte-rich lesions (5 patients with orbital lymphoma; 7 patients with IgG4-ROD). The other 165 (93%) of the 178 patients showed only one lesion in the orbital cavity or eyelid. Sixteen (9%) of the 178 patients had orbital lymphoma (OL) and 9 (5%) patients had IgG4-related orbital disease (IgG4-ROD).

Among these patients, this study only considered patients who had OL or IgG4-ROD with prebiopsy precontrast and postcontrast orbital CT scans, serum IgG4 levels and serum IgG levels and excluded those were younger than 20 years of age or pregnant cases or those lacked prebiopsy CT and serologic data.

As a result, 25 patients pathologically diagnosed as OL (16 patients) or IgG4-ROD (9 patients) met the inclusion criteria and had prebiopsy orbital CT scans, serum IgG4 levels and serum IgG levels. All patients met the eligibility criteria. We enrolled 25 patients to collect and analyze demographic data, symptoms and signs, past medical histories, CT qualitative and quantitative (Hounsfield unit density) features of orbital lesions, serum IgG levels and serum IgG4 levels of patients with IgG4-ROD from those of patients with OL.

### 2.2. CT Imaging Techniques

This study examined orbital CT images taken by a multiple-detector computed tomography (MDCT) scanner for the selected 25 patients. MDCT scanners of orbit or face included iCT 256 (256-slice, *n* = 5), Philips Healthcare, Cleveland, OH, USA, Somatom Sensation 16 (16-slice, *n* = 4), Siemens Healthcare, Forchheim, Germany, ECLOS Hitachi Medical Corporation (16-slice, *n* = 1), Tokyo, Japan, and Aquilion 64, Toshiba Medical Systems (64-slice, *n* = 15), Tochigi, Japan. Orbital CT scans were obtained with or without an intravenous contrast medium, which included iobitridol (Xenetix 350; Guerbet, Rue Jean Chaptal, Aulnary-sous-Bios, France, 350 mg I [Iodine]/mL [milliliter]) and iohexol (Omnipaque 350; GE healthcare, Carrigtohill, Co., Cork, Ireland, 350 mg I/mL). The data records showed that twenty-five patients underwent an intravenous power injection as a bolus of 1.2-mL/kg (kilogram) iodine-based contrast medium at 1 mL/second (s). Postcontrast CT images were performed after the complete injection of contrast medium. The axial sections of precontrast and postcontrast orbital CT images scanned along the transaxial direction with the sections parallel to the optic nerve along a line from the inferior border of the maxillary sinus to the middle part of the frontal region. A Hitachi CT scanner took the slice thickness for image viewing of axial images at 1.25 mm (mm) and other MDCT scanners at 2–4 mm. The coronal and sagittal sections of postcontrast orbital CT images were reformatted with 2–4 mm in slice thickness. The reconstruction matrix for MDCT scans of orbit was 512 × 512.

### 2.3. Analysis of Images and Pathologic Diagnosis

Two experienced radiologists (CHW and YYC) reanalyzed orbital lesions of the 25 patients on orbital CT images with axial, sagittal and/or coronal images together without knowledge of pathologic diagnosis of orbital tumors. The consensus from the two radiologists served as the final interpretation of images. If the two radiologists could not reach an agreement on any features from orbital CT scans, a third experienced radiologist (HCH) mediated the disagreement.

This study analyzed the following orbital CT features of each lesion or of each patient: maximum diameter of a lesion, lesion borders, homogeneity of CT density, a lesion involving extraocular muscle tendons, the lacrimal sac, lacrimal gland, preseptal space, extraconal, conal or intraconal orbital compartments, the optic nerve, infraorbital nerve, presence or absence of bone remodeling, single tumor or multiple lesions and single or bilateral orbital involvement, single or bilateral lacrimal gland involvement. This study also measured the mean values of the precontrast and postcontrast CT Hounsfield unit scales (CTHU) of each orbital lesion among the 25 patients.

A regular border of an orbital tumor on CT scans indicated the contour of a lesion from the surrounding tissue was smooth for more than 75% of the lesion. An irregular border of a lesion showed microlobulated, microangulated or indistinct contour from the surrounding tissue with ≥25% of the lesion. Lesion involvement indicated lesion infiltration, invasion or encasement on orbital CT scans.

This study measured CTHU for all 47 orbital lesions of 25 patients on both pre and postcontrast prebiopsy CT scans. The region of interest (ROI) maker in an oval shape was placed in the center of each lesion to cover 50% of the largest tumor area on CT axial images, avoiding the inclusion of bone and blood vessels (Figure 1). This study also calculated postcontrast CTHU divided by precontrast CTHU.

An experienced pathologist (AFYL) with 29 years of experience in pathology diagnosis reviewed the pathologic and immunohistochemical sections of the specimens of the 25 patients to confirm pathologic results of OL and IgG4-ROD. The two main pathologic criteria of IgG4-ROD included (1) IgG4+/IgG+ ratio > 40%, and/or (2) IgG4+ cells > 10/high-powered field (HPF) in histopathologic examination [5,9,14].

The radiologist (WHY) integrated demographic data, patient symptoms, signs and past histories, prebiopsy serum IgG4 levels and IgG levels and CT imaging interpretations and the mean values of CTHU measurement results of the 25 patients to evaluate the differences in CT qualitative and quantitative features, serum IgG and IgG4 levels between OL and IgG4-ROD.

### 2.4. Statistical Analysis

This study used SPSS version 19.0 software (SPSS, Inc., Chicago, IL, USA) for data analysis. Specially, we applied the Mann–Whitney U test to compare continuous variables because of the small sample size and the χ2 or Fisher’s exact test for categorical variables at the level of significance of *p* < 0.05. Receiver operating characteristic (ROC) curve analysis calculated the area under the ROC curve to identify diagnostic values of CTHU, serum IgG4 levels and serum IgG levels of IgG4-ROD. This study assessed the findings based on sensitivity, specificity and accuracy with a 95% confidence interval (95% CI).

## 3. Results

The median age (mean ± standard deviation [SD], range) of the selected 25 patients was 59 (58.20 ± 10.61, 32–78). The median age (mean ± SD, range) of 16 patients with OL was 60.5 (59.31 ± 9.20, 41–78) and that of 9 patients with IgG4-ROD was 58 (56.22 ± 13.11, 32–69) (*p* = 0.934, Mann–Whitney U test). Of the 25 patients, 17 (68%) were male and 8 (32%) were female. Twelve (12 or 71%) of the 17 male patients were OL and 5 (29%) were IgG4-ROD; four (50%) of 8 females were OL patients and 4 (50%) were IgG4-ROD (*p* = 0.3942, Fisher’s exact test).

The 25 patients showed proptosis, palpable mass and/or eyelid swelling—none of the 25 patients suffered from orbital pain or tender palpable mass. Six (6 or 24%) of the 25 patients had malignancy histories. Five (83%) of the 6 patients with malignant histories had OL: one with renal cell carcinoma, one with prostatic cancer and soft palate follicular lymphoma, one with squamous cell carcinoma of the tongue, one with follicular lymphoma involving lung, neck lymph nodes and bone marrow and one with chronic lymphocytic leukemia. Only one (17%) of the 6 patients with malignant history was an IgG4-ROD patient who had ovarian cancer. Patient malignant histories of the two groups had no significant difference (*p* = 0.3644, Fisher’s exact test).

A pathologist (AFYL) reviewed the pathologic sections of the 25 patients. The pathologic review concluded 13 patients with extranodal marginal zone lymphoma of mucosa-associated lymphoid tissue (MALT lymphoma), 1 with low-grade B cell lymphoma with plasmacytic differentiation, 1 with diffuse large B cell lymphoma, 1 with follicular lymphoma and 9 with IgG4-ROD. The histopathologic findings of the 9 patients with IgG4-ROD showed diffuse lymphoplasmacytic infiltration, IgG4-positive (IgG4+) plasma cells, IgG-positive (IgG+) plasma cells and various degree fibrosis. Seven (78%) of the nine IgG4-ROD patients showed IgG4+ cells > 100 cells/HPF and IgG4+/IgG+ ratio > 40% (Figure 1). Another 2 of the 9 IgG4-ROD patients (22%) had IgG4+ plasma cell < 50 cells/HPF and IgG4+/IgG+ ratio > 40%.

Furthermore, CT images indicated a total of 47 orbital tumors among the 25 patients: 27 lesions were OL and 20 lesions were IgG4-ROD. Of 47 orbital tumors, none appeared inside the eyeball.

Table 1 and Table 2 summarize CT features of 47 tumors among the 25 patients, of which 16 had orbital lymphoma and 9 had IgG4-ROD.

Specifically, of the 16 patients with OL, eleven (69%) had a solitary tumor in an orbital cavity or at eyelids, 1 (6%) had 2 tumors, 2 (13%) had 3 and 2 (13%) had 4. The other 9 out of the 25 patients had IgG4-ROD: 2 (22%) with 1 tumor; 5 (56%) with 2 tumors, 1 (11%) with 3 tumors and 1 (11%) with 5 tumors. CT features statistically significantly associated with IgG4-ROD included lesions with regular borders (*p* = 0.0069), multiple tumors (*p* = 0.0414), lacrimal gland involvement (*p* = 0.0085), lesions involving bilateral lacrimal glands and bilateral orbital cavities (*p* = 0.0022 and *p* = 0.0168, respectively, Figure 1).

In contrast, tumors involving the extraconal, conal or intraconal space, lacrimal sac, optic nerve, extraocular muscle tendon, infraorbital nerve, preseptal space and presence of sinusitis and bone remodeling were ineffectual to differentiate IgG4-ROD from orbital lymphoma (Figure 1 and Figure 2; all *p* > 0.05, Fisher’s exact test). Two IgG4-ROD patients and 6 OL patients had a solitary tumor involving the preseptal space (Figure 2).

Table 3 shows the descriptive statistical prebiopsy values of precontrast CT Hounsfield unit scales (CTHU), postcontrast CTHU and postcontrast CTHU/precontrast CTHU ratios of 27 tumors of OL and 20 tumors of IgG4-ROD on prebiopsy CT scans.

Table 4 demonstrates descriptive statistical prebiopsy values of serum IgG4 levels, serum IgG levels and the ratios of serum IgG4 level/serum IgG level of the 16 patients with OL and the 9 patients with IgG4-ROD.

Figure 3 and Figure 4 show the differences in postcontrast CTHU, postcontrast CTHU/precontrast CTHU ratios, serum IgG4 levels and serum IgG4 level/serum IgG level ratios were statistically significant between the two groups (all *p* < 0.05, Mann–Whitney U test).

Figure 3 shows that the areas under the ROC curve (AUC) of precontrast CTHU, postcontrast CTHU and the ratios of postcontrast CTHU/precontrast CTHU were 0.56 (95% CI: 0.393–0.727, *p* = 0.484), 0.766 (95% CI: 0.615–0.917, *p* = 0.002) and 0.670 (95% CI: 0.498–0.842, *p* = 0.048). According to Figure 4, the AUC for serum IgG4 levels, serum IgG levels and the ratios of serum IgG4/serum IgG were 0.847 (95% CI: 0.674–1.000, *p* = 0.005), 0.684 (95% CI: 0.455–0.913, *p* = 0.134) and 0.819 (95% CI: 0.639–1.000, *p* = 0.009), respectively.

Compared with above data, the AUC using postcontrast CTHU (= 0.766) and serum IgG4 levels (= 0.847) was moderately accurate for the diagnostic yield of IgG4-ROD because both AUC measures fell between 0.7 and 0.9. The largest Jordon index 0.528 [(sensitivity−[1−specificity] = 0.528)] suggested a cutoff value of postcontrast CTHU equal to 89.5. The sensitivity and specificity were 0.75 (95% CI: 0.509–0.913) and 0.778 (95% CI: 0.578–0.914), respectively. As to serum IgG4 level, the largest Jordon index (0.701) suggested a cutoff value equal to 132.5 mg/dL, which resulted in sensitivity of 0.889 (95% CI: 0.518–0.997) and specificity of 0.813 (95% CI: 0.544–0.960), respectively.

For patients with postcontrast CTHU ≥ 89.5 in at least one orbital nodule in the two groups, 7 (58%) were IgG4-ROD patients and 5 (42%) were OL patients (*p* = 0.0414, Fisher’s exact test). For serum IgG4 levels ≥ 132.5 mg/dL, 8 (73%) were IgG4-ROD patients and 3 (27%) OL (*p* = 0.0021).

The postcontrast CTHU and serum IgG4 levels for patients with a solitary orbital tumor in the two groups of OL and IgG4-ROD patients were as follows: higher postcontrast CTHU (≥89.5) in 3 OL patients and 1 IgG4-ROD case; lower postcontrast CTHU (<89.5) in 8 OL and 1 IgG4-ROD; and lower serum IgG4 levels (<132.5) in all 11 OL patients and 1 IgG4-ROD case with higher CTHU. In addition, one IgG4-ROD patient with a lower postcontrast CTHU showed a higher serum IgG4 level ≥ 132.5.

The postcontrast CTHU and serum IgG4 level for patients with multiple orbital lesions in the two groups were as follows: lower postcontrast CTHU (<89.5) were noted in 3 OL patients and in 1 IgG4-ROD case; higher postcontrast CTHU (≥89.5) were found in 2 OL patients and in 6 IgG4-ROD cases; higher serum IgG4 levels (≥132.5 mg/dL) were found in 3 OL patients and in 7 IgG4-ROD cases; lower serum IgG4 levels (<132.5 mg/dL) appeared in 2 OL patients. Two of the three OL patients with lower postcontrast CTHU showed OL involving bilateral lacrimal glands, who had different serum IgG4 levels: 51.3 and 339.5, respectively. Concurrent higher postcontrast CTHU and a higher serum IgG4 level were found in 1 OL patient (1/5, 20%) with multiple tumors in the left orbital cavity and in 6 IgG4-ROD cases with tumors involving bilateral lacrimal glands. Lower postcontrast CTHU and higher serum IgG4 level were noted in only one IgG4-ROD (1/7, 14%) case, who showed tumors mainly in intraconal spaces of bilateral orbits.

If this study used “lesions with bilateral lacrimal gland involvement” (the most significant qualitative CT feature in statistics, *p* = 0.0022), “bilateral lacrimal gland involvement and a higher serum IgG4 level (≥132.5 mg/dL) (*p* = 0.0005)” or “bilateral lacrimal gland involvement and higher postcontrast CTHU (≥89.5, quantitative CT feature) (*p* = 0.00047)” or “higher postcontrast CTHU and a higher serum IgG4 level” (*p* = 0.0029) as helpful test tools for diagnosis of IgG4-ROD (Table 5), sensitivity, specificity and accuracy of the first test (Test 1), the second (Test 2), the third (Test 3) and the latest one (Test 4) were as follows (Table 5): 0.78 (95% CI: 0.3999–0.972), 0.88 (95% CI: 0.617–0.985) and 0.84 (95% CI: 0.639–0.955) for Test 1; 0.78 (95% CI: 0.3999–0.972), 0.94 (95% CI: 0.698–0.998) and 0.88 (95% CI: 0.688–0.975) for Test 2; 0.67 (95% CI: 0.299–0.925), 1 (95% CI: 0.794–1) and 0.88 (95% CI: 0.688–0.975) for Test 3; 0.67 (95% CI: 0.299–0.925), 0.94 (95% CI: 0.698–0.998) and 0.84 (95% CI: 0.639–0.955) for Test 4. In Table 5, Test 3 had 100% of positive predictive value (PPV). Test 1 & Test 2 had the highest negative predictive value (NPV) 0.88.

## 4. Discussion

The IgG4-related disease can result in fibroinflammatory lesions at nearly any anatomic site [14]. OL is malignant and needs radiotherapy, chemotherapy and/or operation [4,7,8]. IgG4-ROD is benign and approximately 90% of patients respond well to steroid treatment [5]. Both of malignant OL and benign IgG4-ROD are lymphoplasmacytic infiltrated mass-like lesions, which make clinicians difficult to differentiate from each other [6]. This study showed that lesions with regular borders, multiple tumors, lacrimal gland involvement, simultaneous involvement of bilateral lacrimal glands and bilateral orbital cavities and higher medians of postcontrast CTHU and serum IgG4 levels were significantly related to IgG4-ROD (all *p* < 0.05). Postcontrast CTHU ≥ 89.5 showed 0.75 sensitivity and 0.778 specificity with the AUC = 0.766 (95% CI: 0.615–0.917, *p* = 0.002); serum IgG4 levels ≥ 132.5 mg/dL had 0.889 sensitivity and 0.813 specificity, with the AUC = 0.847 (95% CI: 0.674–1.000, *p* = 0.005, Figure 3 and Figure 4). A lesion with regular borders is most likely to be a slow growing benign mass or less likely to be an indolent malignant tumor [1]. IgG4-ROD being benign usually presented as lesions with regular borders in this study.

Serum IgG4 levels account for 3% to 6% total amount of serum IgG levels [15]. Hamano et al. [16] reported a cutoff value of 135 mg/dL to differentiate autoimmune pancreatitis from pancreatic cancer with a high sensitivity (95%), specificity (97%) and accuracy (97%). This study identified a cutoff value 132.5 mg/dL (close to 135 mg/dL) to distinguish IgG4-ROD from OL at diagnostic accuracy (AUC) of 84.7% with 88.9% sensitivity and 81.3% specificity.

However, approximately 40–50% of patients with biopsy-proven IgG4-related disease have normal serum IgG4 concentrations [10,14,17]. In our study, normal serum IgG4 (<132.5 mg/dL) occurred in 13 (81%) of 16 OL patients and one IgG4-ROD patient (1/9, 11%), who had a solitary orbital lesion. There may be several reasons to explain why in our study there was a lower percentage of IgG4-ROD with normal serum IgG4 levels: first, our study was a small sample research, which may have selection bias; second, serum IgG4 levels may vary according to the specific organ involved [10]; finally, elevated serum IgG4 levels represent a subtype of IgG4-related disease with more inflammatory features and worsening disease activity [17]. Our IgG4-ROD patients (8/9, 89%) could be developing an active IgG4-related disease with elevated serum IgG4 concentrations.

Patient’s age, standard imaging features and localizing orbital lesions to intraconal, conal or extraconal compartments help limit the differential diagnosis [18]. Our study showed no significant difference in median ages between patients with OL and IgG4-ROD (*p* = 0.934, Mann–Whitney U test). Lesions with regular borders, multiple orbital tumors, lacrimal gland involvement, lesions simultaneously involving bilateral lacrimal glands and bilateral orbital cavities and higher postcontrast CTHU (≥89.5) on orbital CT scans were significantly associated with IgG4-ROD (all *p* < 0.05). The difference in extraconal, conal and intraconal compartments of orbital lesions between OL and IgG4-ROD groups was not statistically significant (*p* = 0.4813). In addition to CT, magnetic resonance imaging (MRI) also helps in further diagnostic workup of orbital tumors and provides ocular anatomy for lesions involvement, perineural spread and intracranial extension [12]. Both retinoblastomas typically found in children and uveal melanomas in adults appear in the globe. Retinoblastoma is slightly hyperintense on T1 weighted MRI (T1WI) and very hypointense relative to vitreous on T2-weighted MRI (T2WI) and well contrast enhancement on postcontrast CT and contrast-enhanced (CE) MRI [12,18]. Ninety percent of retinoblastomas demonstrate calcifications on precontrast CT scans [18]. Melanomas with melanin show characteristic hyperintensity on T1WI and hypointensity on T2WI [12,18]. For intraconal orbital tumors, gliomas common among children result in fusiform enlargement of the optic nerve on axial CT and MRI [12,18]. In contrast, meningiomas, commonly seen in the 5th decade of life, classically show the contrast-enhancing tumor with a “tram-tract” configuration alongside the optic nerve on axial postcontrast CT or CE MRI [12]. The most common benign orbital tumor in adults is a cavernous hemangioma, which typically demonstrates a well-defined dense unilateral orbital intraconal mass with intra-tumoral calcifications on precontrast CT scans and MRI. The enhancement spread pattern on a dynamic postcontrast CT and dynamic CE T1WI can help to distinguish between cavernous hemangioma and schwannoma [18,19]. Cavernous hemangiomas show initial patchy enhancement on arterial phase, but schwannomas start a wide area of enhancement. The most common congenital orbital nodules are dermoids, which usually show a well outlined round or oval tumor with a capsule and low density or fat contents in the extraconal space on CT scans or MRI [18]. Due to fat contents, dermoids typically show hyperintensity on T1WI, hyperintensity on T2WI and hypointensity on short tau inversion recovery MRI (STIR) [18]. Benign mixed tumor of lacrimal gland usually seen in middle-aged patients demonstrates a well-circumscribed round or oval tumor with homogeneous enhancement on postcontrast CT and CE MRI [12]. Malignant epithelial lacrimal gland tumors show a mass with a well- or poor-defined margin with associated bony remodeling or destruction in 70% cases on CT scans [18].

Multiple or multicompartmental orbital masses include venolymphatic malformations (VLM), rhabdomyosarcoma (RMS), plexiform neurofibroma, thyroid ophthalmopathy (TO), orbital pseudotumor (OP), lymphoma, metastases and IgG4-ROD [18]. The first three types of masses are common among children; the last five, among adults [3,5,18]. VLM usually appears poorly defined, lobulated and multiloculated lesions with various signal intensity on T1WI and T2WI [18]. VLM may demonstrate fluid–fluid level on MRI, which is highly suggestive of the diagnosis of VLM [12]. VLM, RMS and plexiform neurofibroma may have similar findings on CT and MRI [18]. TO causes enlarged bilateral myositis of the extraocular muscles, often involves medial and inferior rectus muscles with sparing tendinous insertions on CT and MRI [12] and is related to elevated thyroid-stimulating hormone level [18]. OP, IgG4-ROD and OL show similar MRI features on conventional sequences, which are hypointense on T1WI and T2WI and well contrast enhancement on postcontrast T1WI. Furthermore, diffusion-weighted imaging (DWI) with apparent diffusion coefficient (ADC) mapping can help to differentiate between benign and malignant orbital lesions [1,18]. Sepahdari et al. have reported that an ADC value < 1.0 × 10^−3^ mm^2^/ sec and an ADC ratio < 1.2 are optimal for predicting orbital malignant tumors [1,18]. Prior studies used ADC value < 1.0 × 10^−3^ mm^2^/ sec and ADC ratio < 1.2 to differentiate orbital lymphoma from benign OP and IgG4-ROD with more than 95% accuracy [1,18]. However, ADC values and ratios cannot differentiate OP from IgG4-ROD because the two disease have similar these values [1,18]. OP manifests with the most common acute unilateral painful mass in adults, which assist in differentiating OP from TO, OL and IgG4-ROD. Pain is uncommon in TO, OL and IgG4-ROD [12,18]. None of 25 patients in this study suffered from orbital pain or tender palpable mass, either. Consistent with the result of our research, Fujita et al. have reported that IgG4-ROD commonly presents involving bilateral lacrimal glands [15], which can distinguish OL and OP from IgG4-ROD. A clinician can suggest the diagnosis of orbital metastasis only when clinically primary malignancy is known [18].

An effective clinical diagnosis or appropriate disease classification for IgG4-related disease needs the integration of clinical findings, radiological features and serologic or pathologic data [12]. None of our 25 patients suffered from painful orbital lesions. Four tests for IgG4-ROD diagnosis used in this study included serologic serum IgG4 levels, radiological CT qualitative (lesions with bilateral lacrimal gland involvement) and/or quantitative features (postcontrast CTHU ≥ 89.5). Of the four tests, Test 2 (lesions with bilateral lacrimal gland involvement and a higher serum IgG4 level [≥132.5 mg/dL]) with the highest sensitivity (78%), a higher specificity (94%), a higher PPV (88%), the highest NPV (88%) and the highest accuracy (88%) could be the better prebiopsy test to distinguish IgG4-ROD from OL.

This study had only two IgG4-ROD patients with a solitary tumor, which was a small sample and lacked specific CT features. Prebiopsy diagnosis of a solitary IgG4-ROD could depend on a painless orbital mass, postcontrast CTHU ≥ 89.5 and serum IgG4 level ≥ 132.5 mg/dL. Tissue proof is an ultimate diagnostic way. However, biopsy is not always suitable for orbital lesions. The best medical option for a benign mixed tumor or malignant mass of the lacrimal gland may be excision en bloc without biopsy once clinical and imaging diagnosis. However, incomplete excision or ruptures of neoplasms via biopsy may result in tumor recurrence, malignant transformation of a mixed tumor and poor prognosis [12]. In clinical practice, some IgG4-ROD patients may be at high risk for biopsy and/or refuse biopsy. However, once these patients meet possible diagnosis of IgG4-ROD [20], systemic steroid treatment may be a good alternative. The criteria for possible IgG4-ROD diagnosis include elevated serum IgG4 (≥135 mg/dL), enlargement of the lacrimal gland or masses, enlargement or hypertrophic lesions in various orbital tissues [20]. Clinicians could forgo further biopsy if such patients respond well to glucocorticoids within weeks, such as reductions in the size of tumors, improvements of symptoms and a significant decrease in serum IgG4 [20]. Alternative non-vital organ or lip biopsy may be an acceptable option [21].

Sato et al. [22] reported that 17 (81%) of 21 patients with IgG4-ROD had involvement of the lacrimal glands and 13 (70.6%) of 17 cases showed bilateral lacrimal gland swellings. Our study also showed that seven (78%) of nine patients with IgG4-ROD had bilateral lacrimal gland involvement, which could distinguish IgG4-ROD from OL (*p* = 0.0022). Neither Sato et al. nor this study had patients with IgG4-ROD originating from conjunctival or subconjunctival tissue. However, prior research suggested that IgG4-ROD can also develop at conjunctival tissue [23,24].

This study has several limitations. First, this retrospective research had a small sample of patients. Second, the patients in this study received different brands of contrast agents and CT machine, which could produce potential bias in measurement of CTHU. Last, incomplete data records ruled out the possibility of deciphering a detailed correlation between CT features and clinical presentations of OL and IgG4 ROD.

## 5. Conclusions

This study compared prebiopsy precontrast and postcontrast CT features, serum IgG4 and serum IgG levels of IgG4-ROD with those of OL. The key findings showed that IgG4-ROD had high correlation with the presence of lesions with regular borders, presence of multiple lesions, lesions involving the lacrimal gland, both lacrimal glands and bilateral orbital cavities on CT scans, higher values of postcontrast CTHU, postcontrast CTHU/precontrast CTHU ratios, serum IgG4 levels and serum IgG4/IgG level ratios (all *p* < 0.05). For diagnosis of IgG4-ROD, postcontrast CTHU ≥ 89.5 and serum IgG4 level ≥ 132.5 mg/dL provided moderate diagnostic accuracy, AUC = 0.776 and 0.847, respectively, which were higher than those of postcontrast HU/precontrast HU and serum IgG4/IgG level ratio. The special CT features and elevated serum IgG4 levels could help differentiate IgG4-ROD from OL. Prebiopsy diagnosis of the uncommon solitary type of IgG4-ROD could also depend on a painless orbital mass and elevated CTHU ≥ 89.5 and serum IgG4 level ≥ 132.5 mg/dL.

## Figures and Tables

**Figure 1 jcm-09-02425-f001:**
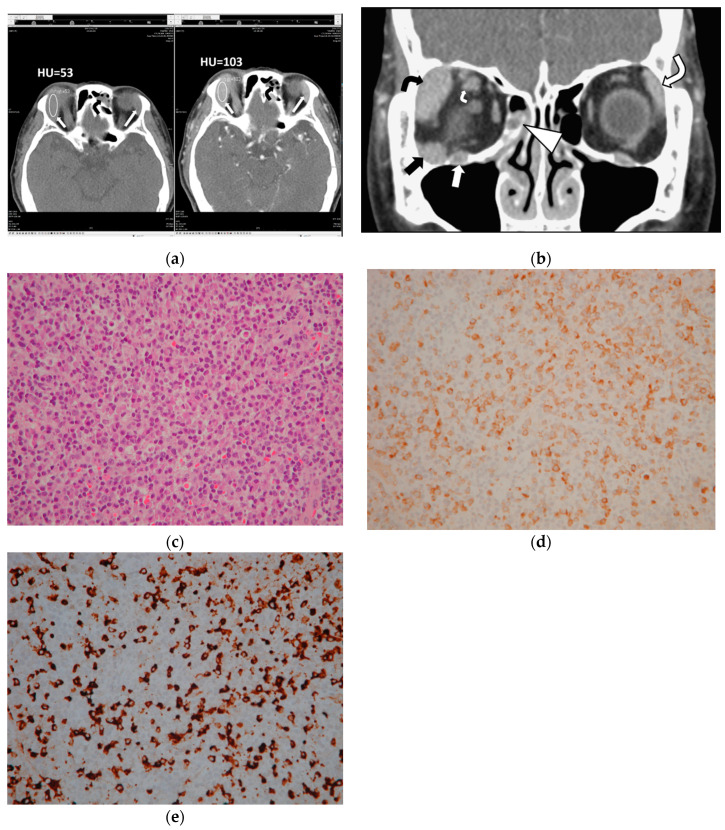
A 68-year-old man with immunoglobulin G4 (IgG4)-related orbital disease (IgG4-ROD) shows multiple tumors in bilateral orbital cavities. (**a**) Axial computed tomography (CT) scans show masses in bilateral lacrimal glands (short and large arrows). The mean value of CT Hounsfield unit scale (CTHU) is measured at the enlarged right lacrimal gland (short arrows) on a picture archiving and communication system monitor. The region of interest (ROI) marker in an oval shape is placed in the center of the mass (short arrows) to cover 50% of the largest tumor area. The mean value of precontrast CTHU is 53 and that of postcontrast CTHU is 103. Sinusitis is found in the left frontal sinus with mucus retention (black curved arrows); (**b**) Coronal postcontrast CT scan shows multiple masses or enlargement in various ophthalmic tissues with regular borders and homogeneous contrast enhancement in bilateral orbital cavities as follows: a mass at the extraconal compartment of the right orbital cavity (black arrow),the right lacrimal gland (black curved arrow), the left lacrimal gland (large white curve arrow), the right superior rectus muscle belly (small white curve arrow) and the right infraorbital nerve (white arrow). Sinusitis is noted in the right ethmoid sinus with mucus retention (arrowhead); (**c**) Pathologic specimen shows infiltration of many lymphoplasma cells and mild fibrosis (hematoxylin–eosin stain, original magnification ×200); (**d**) Immunostaining for immunoglobulin G (IgG)-expression shows many plasma cells are positive for IgG stains (original magnification ×200); (**e**) Immunostaining for IgG4-expression shows abundant IgG4-positive plasma cells have infiltrated the lesion. IgG4-postive/IgG-positive plasma cell ratio is more than 40%. There are more than 100 IgG4-positive plasma cells in one high-powered field (>100/HPF) (original magnification ×200).

**Figure 2 jcm-09-02425-f002:**
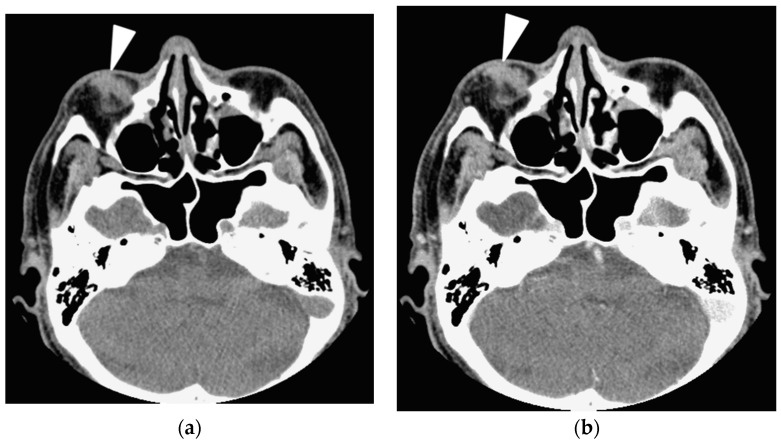
A 66-year-old man with orbital lymphoma shows a painless solitary lesion at the right lower eyelid. (**a**) Axial precontrast CT image shows a small nodule with an irregular border and homogeneous density involves the preseptal space of the right lower eyelid (arrowhead). Precontrast CT Hounsfield unit scale (CTHU) of the nodule is 57; (**b**) Axial postcontrast CT image shows the nodule demonstrates homogeneous enhancement (arrowhead). Postcontrast CTHU of the nodule is 68. Serum IgG4 level of the patient is 44.3 mg/dL.

**Figure 3 jcm-09-02425-f003:**
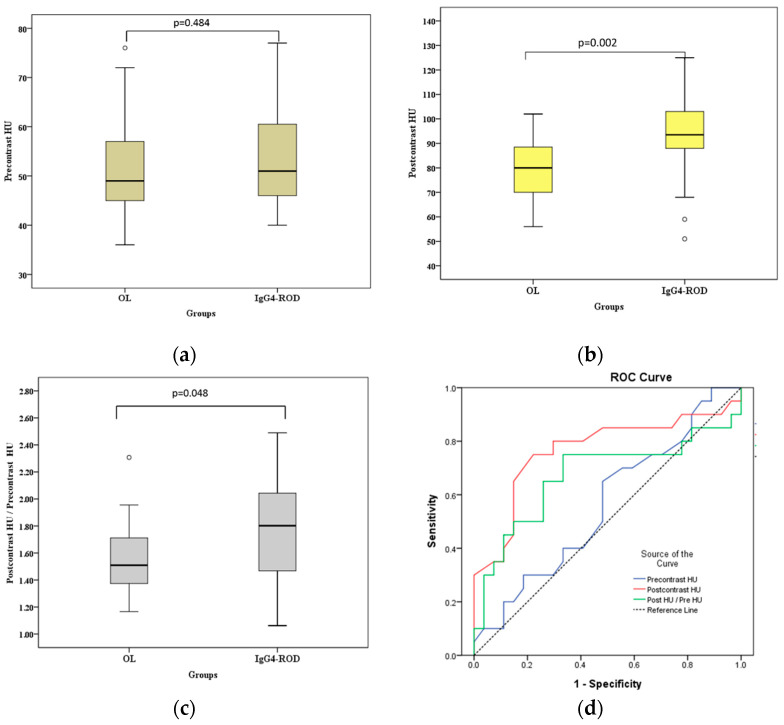
Box-and-whisker plots and receiver operating characteristic (ROC) curve analysis of precontrast computed tomography (CT) Hounsfield unit scale (CTHU), postcontrast CTHU, postcontrast CTHU/precontrast CTHU ratios between orbital lymphoma (OL) and immunoglobulin G4-related orbital disease (IgG4-ROD). Bars = medians. (**a**) Precontrast CTHU (precontrast HU) shows a nonsignificant difference between the two groups (*p* = 0.484, Mann–Whitney U test); (**b**) Postcontrast CTHU (postcontrast HU) shows a significant difference between the two groups (*p* = 0.002, Mann–Whitney U test); (**c**) Postcontrast CTHU/precontrast CTHU ratios (postcontrast HU/precontrast HU) show a significant difference between the two groups (*p* = 0.048, Mann–Whitney U test); (**d**) Areas under the ROC curve (AUC) of precontrast CTHU (precontrast HU), postcontast CTHU (postcontrast HU) and postcontrast CTHU/precontrast CTHU ratios (postcontrast HU/precontrast HU) are 0.56 (95% CI: 0.393–0.727, *p* = 0.484), 0.766 (95% CI: 0.615–0.917, *p* = 0.002) and 0.670 (95% CI: 0.498–0.842, *p* = 0.048), respectively. ○—an outlier.

**Figure 4 jcm-09-02425-f004:**
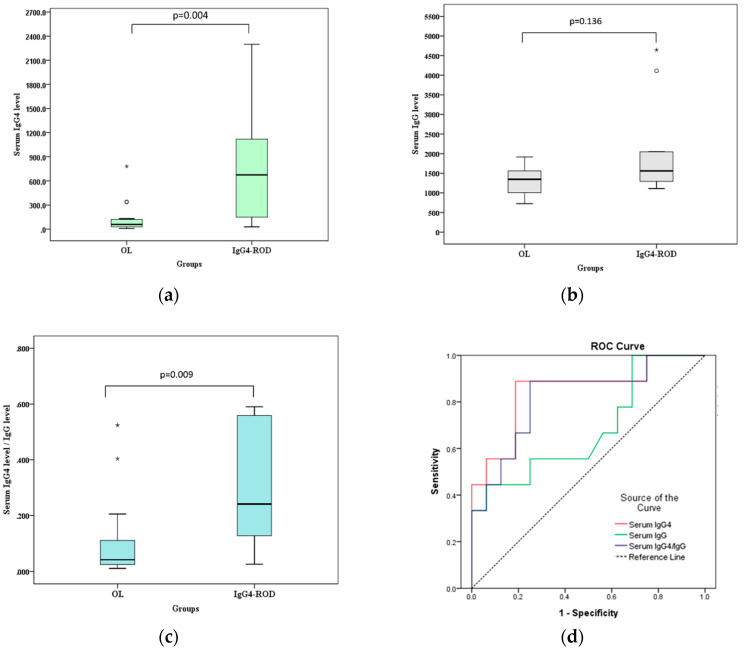
Box-and-whisker plots and receiver operating characteristic curve (ROC) analysis of serum immunoglobulin G4 (IgG4) levels, serum immunoglobulin G (IgG) levels and serum IgG4 level/serum IgG level ratios between orbital lymphoma (OL) and IgG4-related orbital disease (IgG4-ROD). Bars = medians. (**a**) Serum IgG4-levels demonstrate a significant difference between the two groups (*p* = 0.004, Mann–Whitney U test); (**b**) Serum IgG levels demonstrate a nonsignificant difference between the two groups (*p* = 0.136, Mann–Whitney U test); (**c**) Serum IgG4 level/IgG level ratios (serum IgG4 level/IgG level) demonstrate a significant difference between the two groups (*p* = 0.009, Mann–Whitney U test); (**d**) The areas under the ROC curve of serum IgG4 level (serum IgG4), serum IgG level (serum IgG) and serum IgG4 level/IgG level ratio (serum IgG4/IgG) are 0.847 (95% CI: 0.674 to 1.000, *p* = 0.005), 0.684 (95% CI: 0.455 to 0.913, *p* = 0.134) and 0.819 (95% CI: 0.639 to 1.000, *p* = 0.009), respectively. * and ○—outliers.

**Table 1 jcm-09-02425-t001:** Computed tomography (CT) features of 47 tumors among the 25 patients with orbital lymphoma or immunoglobulin G4-related orbital disease (IgG4-ROD) on prebiopsy orbital CT scans.

Orbital CT Scans	Orbital Lymphoma *n* (%)	IgG4-ROD *n* (%)	*p*
**Tumor Size, median (mean ± SD, range)**	2.58 (2.62 ± 1.149, 0.98–5.16)	3.17 (2.70 ± 1.233, 0.66–5.1)	0.667 ^@^
**Lesion border**			0.0069 ^#^
Regular	10 (38)	16 (62)	
Irregular	17 (81)	4 (19)	
**Precontrast CT density**			1 ^#^
Homogeneous	26 (57)	20 (43)	
Heterogeneous	1 (100)	0 (0)	
**Postcontrast CT contrast-enhancement**			1 ^#^
Homogeneous	26 (57)	20 (43)	
Heterogeneous	1 (100)	0 (0)	
**Extraocular muscle tendon involvement**			0.1138 ^#^
Presence	10 (77)	3 (23)	
Absence	17 (50)	17 (50)	
**Lacrimal sac involvement**			1 ^#^
Presence	3 (60)	2 (40)	
Absence	24 (57)	18 (43)	
**Preseptal space involvement**			1 ^#^
Presence	8 (62)	5 (38)	
Absence	19 (56)	15 (44)	
**Lacrimal gland involvement**			0.0085 ^#^
Presence	8 (36)	14 (64)	
Absence	19 (76)	6 (24)	
**Orbital compartment involvement**			0.4813 ^#^
Extraconal or/and conal	20 (53)	17 (47)	
Intraconal	7 (78)	3 (22)	
**Optic nerve involvement**			1 ^#^
Presence	5 (63)	3 (37)	
Absence	22 (56)	17 (44)	
**Infraorbital nerve involvement**			1 ^#^
Presence	1 (50)	1 (50)	
Absence	26 (58)	19 (42)	
**Bone remodeling**			0.2507 ^#^
Presence	3 (100)	0 (0)	
Absence	24 (55)	20 (45)	

*n* (%)—number (percentage); SD—standard deviation; ^@^—Mann–Whitney U test; ^#^—Fisher’s exact test.

**Table 2 jcm-09-02425-t002:** CT features of the 25 patients with orbital lymphoma or immunoglobulin G4-related orbital disease (IgG4-ROD) on prebiopsy orbital CT scans.

Orbital CT Scans	Orbital Lymphoma *n* (%)	IgG4-ROD *n* (%)	*p*
**Tumor number**			0.0414 ^#^
Single	11 (85)	2 (15)	
Multiple (≥2)	5 (42)	7 (58)	
**Orbital involvement**			0.0168 ^#^
One side	12 (86)	2 (14)	
Bilateral	4 (36)	7 (64)	
**Bilateral lacrimal gland involvement**			0.0022 ^#^
Presence	2 (22)	7 (78)	
Absence	14 (88)	2 (12)	
**Sinusitis**			0.6882 ^#^
Presence	7 (58)	5 (42)	
Absence	9 (69)	4 (31)	

*n* (%)—number (percentage); ^#^—Fisher’s exact test.

**Table 3 jcm-09-02425-t003:** Descriptive statistical prebiopsy values of precontrast CT Hounsfield unit scales (Pre HU), postcontrast CTHU (Post HU) and postcontrast CTHU/precontrast CTHU ratios (Post HU/Pre HU) of 27 tumors of orbital lymphoma and 20 tumors of IgG4-related orbital disease (IgG4-ROD) on prebiopsy orbital CT scans.

	Orbital Lymphoma *n* = 27			IgG4-ROD *n* = 20		
Parameter	Pre HU	Post HU	Post HU/Pre HU	Pre HU	Post HU	Post HU/Pre HU
Median	49	80	1.51	51	93.5	1.8
Mean	51.8	78.9	1.55	54.1	93.8	1.77
SD	10.28	13.58	0.27	10.69	19.02	0.42
Quartile 1	45	68	1.37	45.5	87	1.41
Quartile 3	57	89	1.76	61.25	103	2.08
minimum	36	56	1.17	40	51	1.06
Maximum	76	102	2.31	77	125	2.49
Outlier 1	76		2.31		51	
Outlier 2					59	

*n*—number; SD—standard deviation.

**Table 4 jcm-09-02425-t004:** The descriptive statistical prebiopsy values of serum IgG4 levels (serum IgG4), serum immunoglobulin G (IgG) levels (serum IgG) and the ratios of serum IgG4 level/serum IgG level (serum IgG4/IgG) of the 16 patients with orbital lymphoma and the 9 patients with IgG4-ROD.

	Orbital Lymphoma *n* = 16			IgG4-ROD *n* = 9		
Parameter (mg/dL)	Serum IgG4	Serum IgG	Serum IgG4/IgG	Serum IgG4	Serum IgG	Serum IgG4/IgG
Median	57.65	1345	0.042	675	1560	0.241
Mean	135.73	1307.94	0.106	756.49	2108.56	0.32
SD	199.82	332.42	0.1502	733.78	1326.07	0.2325
Quartile 1	28.23	979	0.022	142	1232	0.107
Quartile 3	125.4	1594.5	0.123	1140	3080	0.563
minimum	8.1	725	0.011	28.9	1110	0.026
Maximum	780	1915	0.524	2297.9	4645	0.59
Outlier 1	339.5		0.404		4112	
Outlier 2	780		0.524		4645	

*n*—number; mg/dL—milligram/deciliter; SD—standard deviation.

**Table 5 jcm-09-02425-t005:** Contingency table of four helpful testing tools for diagnosis of IgG4-related orbital disease (IgG4-ROD).

	Test 1	Test 2	Test 3	Test 4
True positive, *n*	7	7	6	6
False negative, *n*	2	2	3	3
False positive, *n*	2	1	0	1
True negative, *n*	14	15	16	15
Sensitivity (95% CI)	0.78 (0.40–0.972)	0.78 (0.40–0.972)	0.67 (0.299–0.925)	0.67 (0.299–0.925)
Specificity (95% CI)	0.88 (0.617–0.985)	0.94 (0.698–0.998)	1.0 (0.794–1.0)	0.94 (0.698–0.998)
PPV (95% CI)	0.78 (0.478–0.931)	0.88 (0.504–0.98)	1.0 (*)	0.86 (0.460–0.977)
NPV (95% CI)	0.88 (0.670–0.960)	0.88 (0.687–0.962)	0.84 (0.679–0.931)	0.83 (0.663–0.927)
Accuracy (95% CI)	0.84 (0.639–0.955)	0.88 (0.688–0.975)	0.88 (0.688–0.975)	0.84 (0.639–0.955)

*n*, patient number; Test 1 to Test 4 represent four helpful tools for diagnosis of IgG4-ROD; Test 1, orbital lesions with bilateral lacrimal gland involvement on CT scans; Test 2, orbital lesions with bilateral lacrimal gland involvement on CT scans and a higher serum IgG4 level (≥132.5 mg/dL [milligrams per deciliter]); Test 3, orbital lesions with bilateral lacrimal gland involvement and higher postcontrast CTHU (CT Hounsfield unit scales ≥ 89.5) on CT scans; Test 4, orbital lesions with higher postcontrast CTHU (≥89.5) and a higher serum IgG4 level (≥132.5 mg/dL); (95% CI), (95% confidence interval); *—not shown in the statistics operation; PPV—positive predictive value—NPV—negative predictive value.

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
