# Peer review of "Evaluate the Differences in CT Features and Serum IgG4 Levels between Lymphoma and Immunoglobulin G4-Related Disease of the Orbit"

_jcm, 2020, doi:10.3390/jcm9082425_

Round 1
Reviewer 1 Report
First of all, I would like to congratulate you for the great work you have done. I have a few comments.
This is a well-designed study that attempts to identify markers that can non-invasively differentiate OL from IgG4-ROD. It concludes that certain markers such as IgG4 and IgG4/IgG ratio and CT findings are helpful with moderate accuracy
General comments:
The manuscript needs English review including the use of definite and indefinite articles. There are several grammatical and sentence-building errors including the use of the wrong tense (e.g. many sentences in the discussion)
You used IgG4-ROD and IgG4-RD. Use one abbreviation to make it consistent
Introduction:
Paragraph 2: Lymphoma of the orbital cavity does not always require radiation, chemotherapy and surgery. Instead of "and", I suggest using "and/or".
Paragraph 4: lymphoma is not treated by surgical excision. The purpose of surgery is often to get biopsy for diagnosis. Therefore, the sentence that incomplete excision leads to recurrence is incorrect (does apply to lymphoma). Please correct.
Material and methods:
Paragraph 1: "Approved" not "proved".
Some of the abbreviations was not described at first mention (e.g. TVGH)
Paragraph 4:It was mentioned that none of the patients were pregnant or <20year-old. This is already mentioned under exclusion criteria and redundant. You can just mention that all patients met the eligibility criteria
In the last paragraph of page 5 when mentioning that IgG4-ROS, use an ascending order to list the number of tumors (3 tumors should be listed before 5)
In page 13 last paragraph before discussion, the conclusion that these tests are diagnostic is not supported by the reported figures. Positive predictive values (PPV) are not reported either, which are usually helpful is assessing the predictive value of the tests. Also, the accuracy of all of these tests are far from optimal. Therefore, I would suggest adding PPV if available. You may revise the paragraph to mention that these tests are helpful tool for diagnosis rather than diagnostic. A table to summarize the sensitivity, specificity, PPV, NPV and accuracy for all tested variable would be very helpful.
Discussion:
In discussion paragraph 4, explain why in your study there is a lower percentage of IgG4-ROD with normal IgG4 compared to other reports.
In discussion, paragraph 4 is confusing. It seems that you try to say that high IgG4 is associated with higher inflammation in both IgG4-ROD and OL but showing that this is more in IgG4-ROD. If this is correct, I would suggest starting by linking IgG4 level to inflammation with citation, list the percentage if IgG4 negative OL and IgG4-ROD in this study and literature, link IgG4 to IgG4-ROD based on your figures, discuss the value in differentiating OL from IgG4-ROD and finally can discuss the association between IgG4 and the multiplicity of the tumors (something like that)
In discussion paragraph 5, you mentioned the characteristics of many tumors. Elaborate after the first sentence a little more on the distinguishing features of OL and IgG4-ROD (age, CT characteristics, etc)
Figures:
Figure 1b is blurry
The legends are very long, particularly figure 1. Please be more concise
There is no need for 4 different CT figures, all are somewhat similar to each other. I would suggest using only 2 figures (one for a solitary tumor and one for multiple tumors)
Best of luck..
Author Response
Review Report Form
Open Review
( ) I would not like to sign my review report
(x) I would like to sign my review report
English language and style
(x) Extensive editing of English language and style required
( ) Moderate English changes required
( ) English language and style are fine/minor spell check required
( ) I don't feel qualified to judge about the English language and style
|
Yes |
Can be improved |
Must be improved |
Not applicable |
|
|
Does the introduction provide sufficient background and include all relevant references? |
(x) |
( ) |
( ) |
( ) |
|
Is the research design appropriate? |
(x) |
( ) |
( ) |
( ) |
|
Are the methods adequately described? |
(x) |
( ) |
( ) |
( ) |
|
Are the results clearly presented? |
(x) |
( ) |
( ) |
( ) |
|
Are the conclusions supported by the results? |
(x) |
( ) |
( ) |
( ) |
Comments and Suggestions for Authors
First of all, I would like to congratulate you for the great work you have done. I have a few comments.
This is a well-designed study that attempts to identify markers that can non-invasively differentiate OL from IgG4-ROD. It concludes that certain markers such as IgG4 and IgG4/IgG ratio and CT findings are helpful with moderate accuracy
Reply: Thank you for your comments.
General comments:
The manuscript needs English review including the use of definite and indefinite articles. There are several grammatical and sentence-building errors including the use of the wrong tense (e.g. many sentences in the discussion)
You used IgG4-ROD and IgG4-RD. Use one abbreviation to make it consistent
Reply: Thank you for your comments. An English editor has reviewed this manuscript again and revised many errors including the use of definite and indefinite articles, grammatical and sentence-building errors and the wrong tense. We exchanged IgG4-RD into IgG4-related disease in this manuscript.
Introduction:
Paragraph 2: Lymphoma of the orbital cavity does not always require radiation, chemotherapy and surgery. Instead of "and", I suggest using "and/or".
Reply: We exchanged “and” into “and/or” in Page 1 line 17, Page 2 line 47 and Page 12 line 313 with marks in yellow color (clean copy)
Page 1 line 17(clean copy): radiotherapy and/or surgery of OL. The objective of this study was to report the differences in CT
Page 2 line 47(clean copy): radiotherapy, systemic chemotherapy and/or surgery [7, 8].
Page 12 line 313(clean copy): OL is malignant and needs radiotherapy, chemotherapy and/or operation [4, 7, 8].
Paragraph 4: lymphoma is not treated by surgical excision. The purpose of surgery is often to get biopsy for diagnosis. Therefore, the sentence that incomplete excision leads to recurrence is incorrect (does apply to lymphoma). Please correct.
Reply: Thank you for your advice.
We exchange the incorrect sentence “orbital neoplasm rupture via biopsy or incomplete excision could lead to tumor recurrence and poor prognosis” into “orbital neoplasm rupture via biopsy could may lead to tumor seeding and poor prognosis” in Page 2 line 56-57 with marks in yellow color (clean copy). Furthermore, the sentence dose not mean orbital lymphoma alone.
Page 2 line 56-67(clean copy)
orbital neoplasm rupture via biopsy could may lead to tumor seeding and poor prognosis
Material and methods:
Paragraph 1: "Approved" not "proved".
Reply: We exchanged “proved” into “approved” in Page 2, line 64 with marks in yellow color (clean copy)
Page 2 line 64: (clean copy)
The Institutional Review Board of Taipei Veterans General Hospital (TVGH) approved this
Some of the abbreviations was not described at first mention (e.g. TVGH)
Reply: Thank you for your comment. TVGH is the abbreviation of Taipei Veterans General Hospital. We add TVGH at the end of “Taipei Veterans General Hospital” in Page 2 line 64 with marks in yellow color (clean copy)
Page 2 line 64: (clean copy)
The Institutional Review Board of Taipei Veterans General Hospital (TVGH) approved this
Paragraph 4:It was mentioned that none of the patients were pregnant or <20year-old. This is already mentioned under exclusion criteria and redundant. You can just mention that all patients met the eligibility criteria
Reply: Thank you. We exchange “None of 25 patients were pregnant or younger than 20 years of age.” Into “All patients met the eligibility criteria” in Page 2 line 82 with marks in yellow color (clean copy).
Page 2 line 82: (clean copy)
All patients met the eligibility criteria.
In the last paragraph of page 5 when mentioning that IgG4-ROS, use an ascending order to list the number of tumors (3 tumors should be listed before 5)
Reply: We used ascending order to list the number of tumors. We exchange “5 (56%) of 9 with 2 tumors, 1 (11%) with 5, and 1 (11%) with 3.” into “5 (56%) with 2 tumors, 1 (11%) with 3 tumors, and 1 (11%) with 5 tumors.” in Page 6 line 179-180 with marks in yellow color (clean copy).
Page 6 line 179-180: (clean copy)
had IgG4-ROD: 2 (22%) with 1 tumor; 5 (56%) with 2 tumors, 1 (11%) with 3 tumors, and 1 (11%) with 5 tumors.
In page 13 last paragraph before discussion, the conclusion that these tests are diagnostic is not supported by the reported figures. Positive predictive values (PPV) are not reported either, which are usually helpful is assessing the predictive value of the tests. Also, the accuracy of all of these tests are far from optimal. Therefore, I would suggest adding PPV if available. You may revise the paragraph to mention that these tests are helpful tool for diagnosis rather than diagnostic. A table to summarize the sensitivity, specificity, PPV, NPV and accuracy for all tested variable would be very helpful.
Reply: Thank you for your comments.
1) We made a table (Table 5) to summarize the true positive number (n), true negative n, false positive n, false negative n, sensitivity, specificity, PPV, NPV and accuracy of Test 1-Test 4 in Page 11-12, line 302-309 with marks in yellow color (clean copy).
Table 5. Contigency table of four helpful testing tools for diagnosis of IgG4-related orbital disease (IgG4-ROD).
|
|
Test 1 |
Test 2 |
Test 3 |
Test 4 |
|
True positive, n |
7 |
7 |
6 |
6 |
|
False negative, n |
2 |
2 |
3 |
3 |
|
False positive, n |
2 |
1 |
0 |
1 |
|
True negative, n |
14 |
15 |
16 |
15 |
|
Sensitivity (95% CI) |
0.78 (0.40-0.972) |
0.78 (0.40-0.972) |
0.67 (0.299-0.925) |
0.67 (0.299-0.925) |
|
Specificity (95% CI) |
0.88 (0.617-0.985) |
0.94 (0.698-0.998) |
1.0 (0.794-1.0) |
0.94 (0.698-0.998) |
|
PPV (95% CI) |
0.78 (0.478-0.931) |
0.88 (0.504-0.98) |
1.0 (*) |
0.86 (0.460-0.977) |
|
NPV (95% CI) |
0.88 (0.670-0.960) |
0.88 (0.687-0.962) |
0.84 (0.679-0.931) |
0.83 (0.663-0.927) |
|
Accuracy (95% CI) |
0.84 (0.639-0.955) |
0.88 (0.688-0.975) |
0.88 (0.688-0.975) |
0.84 (0.639-0.955) |
n, patient number; Test 1 to Test 4 represent four helpful tools for diagnosis of IgG4-ROD. Test 1, orbital lesions with bilateral lacrimal gland involvement on CT scans; Test 2, orbital lesions with bilateral lacrimal gland involvement on CT scans and a higher serum IgG4 level (≥132.5 mg/dl); Test 3, orbital lesions with bilateral lacrimal gland involvement and higher postcontrast CTHU (CT Hounsfield unit scales ≥ 89.5) on CT scans; Test 4, orbital lesions with higher postcontrast CTHU (≥ 89.5) and a higher serum IgG4 level (≥132.5 mg/dl); (95% CI), (95% confidence interval); *, not shown in the statistics operation; PPV, positive predictive value, NPV, negative predictive value
2) We exchanged “as diagnostic tests for IgG4-ROD” into “as helpful testing tools for diagnosis of IgG4-ROD (Table 5),” in Page 11, line 294 with marks in yellow color (clean copy).
Page 11 line 294: (clean copy)
a higher serum IgG4 level” (p = 0.0029) as helpful test tools for diagnosis of IgG4-ROD (Table 5),
3) We also added the description” In Table 5, Test 3 had 100% of positive predictive value (PPV). Test 1 & Test 2 had the highest negative predictive value (NPV) 0.88.” in Page 11, line300-301 with marks in yellow color (clean copy).
Page 11 line 300-301: (clean copy)
In Table 5, Test 3 had 100% of positive predictive value (PPV). Test 1 & Test 2 had the highest negative predictive value (NPV) 0.88.
Discussion:
In discussion paragraph 4, explain why in your study there is a lower percentage of IgG4-ROD with normal IgG4 compared to other reports.
Reply: Thank you for your comments. We explain the reason in the paragraph in Page 12, line 331-338 with marks in yellow color.
Page 12, line 331-338(clean copy)
In our study, normal serum IgG4 (<132.5 mg/dL) occurred in 13 (81%) of 16 OL patients and one IgG4-ROD patient (1/9, 11%), who had a solitary orbital lesion. There may be several reasons to explain why in our study there was a lower percentage of IgG4-ROD with normal serum IgG4 levels: first, our study was a small sample research, which may have selection bias; second, serum IgG4 levels may vary according to the specific organ involved [10]; finally, elevated serum IgG4 levels represent a subtype of IgG4-related disease with more inflammatory features and worsening disease activity [17]. Our IgG4-ROD patients (8/9, 89%) could be developing an active IgG4-related disease with elevated serum IgG4 concentrations.
In discussion, paragraph 4 is confusing. It seems that you try to say that high IgG4 is associated with higher inflammation in both IgG4-ROD and OL but showing that this is more in IgG4-ROD. If this is correct, I would suggest starting by linking IgG4 level to inflammation with citation, list the percentage if IgG4 negative OL and IgG4-ROD in this study and literature, link IgG4 to IgG4-ROD based on your figures, discuss the value in differentiating OL from IgG4-ROD and finally can discuss the association between IgG4 and the multiplicity of the tumors (something like that)
Reply: Thank you for your comments. We deleted the descriptions:” Katsura et al. reported that 14 (93%) of 15 patients with neck, head and brain had elevated serum IgG4 concentration (>135 mg/dl). This study proposed that multiple nodules of IgG4-ROD and OL could tend to more inflammatory condition.” We gave up discussing this point because it is not our research goal.
In discussion paragraph 5, you mentioned the characteristics of many tumors. Elaborate after the first sentence a little more on the distinguishing features of OL and IgG4-ROD (age, CT characteristics, etc)
Reply: Thank you for your comments. We elaborated after the first sentence on the distinguishing features of OL and IgG4-ROD (age, CT characteristics, etc) in Page 12 line 340-346 with marks in yellow color. Moreover, we also elaborate the distinguishing MRI features of OL and IgG4-ROD in Page 13, line 382-386 with marks in blue color
Page 12 line 340-346(clean copy):
Our study showed no significant difference in median ages between patients with OL and IgG4-ROD (p = 0.934, Mann-Whitney U test). Lesions with regular borders, multiple orbital tumors, lacrimal gland involvement, simultaneously involving bilateral lacrimal glands and bilateral orbital cavities, and higher postcontrast CTHU (≥ 89.5) on orbital CT scans were significantly associated to IgG4-ROD (all p < 0.05). The difference in extraconal, conal, and intraconal compartments of orbital lesions between OL and IgG4-ROD groups was not statistically significant (p = 0.4813).
Page 13 line 380-388(clean copy):
OP, IgG4-ROD and OL show similar MRI features on conventional sequences, which are hypointense on T1WI and T2WI and well contrast enhancement on postcontrast T1WI. Furthermore, diffusion-weighted imaging (DWI) with apparent diffusion coefficient (ADC) mapping can help to differentiate between benign and malignant orbital lesions [1, 18]. Sepahdari et al have reported that an ADC value < 1.0x 10-3 mm2/ sec and an ADC ratio < 1.2 are optimal for predicting orbital malignant tumors [1, 18]. Prior studies used ADC value < 1.0x 10-3 mm2/ sec and ADC ratio < 1.2 to differentiate orbital lymphoma from benign OP and IgG4-ROD with more than 95% accuracy [1, 18]. However, ADC values and ratios cannot differentiate OP from IgG4-ROD because the two disease have similar these values [1, 18].
Figures:
Figure 1b is blurry
The legends are very long, particularly figure 1. Please be more concise
There is no need for 4 different CT figures, all are somewhat similar to each other. I would suggest using only 2 figures (one for a solitary tumor and one for multiple tumors)
Best of luck.
Reply: Thank your for your advice.
- Figure 1b is blurry. We deleted the blurry Figure 1b and replace a new clear Figure 1b in Page 6 line 184 (clean copy)
Page 6 line 184 (clean copy), a new clear Figure 1b
(Please see attachment)
- We removed some descriptions in the legends of Figure 1 and Figure 2 to make them more concise in page 7 line 185-203 and page 7 line 210-215 with marks in yellow color (clean copy). We removed prior Figure 2 and Figure 3, and we used only prior Figure 1 and prior Figure 4 as Figure 1 and Figure 2 (Page6 line 184 and Page 7 line 209, clean copy);
Page 6-7 line 184-203 and Page 7 line 209-215 (clean copy)
Figure 1:
(Please see attachment)
Figure 1. A 68-year-old man with IgG4-related orbital disease (IgG4-ROD) shows multiple tumors in bilateral orbital cavities. (a) Axial CT scans show masses in bilateral lacrimal glands (short and large arrows). The mean value of CT Hounsfield unit scale (CTHU) is measured at the enlarged right lacrimal gland (short arrows) on a picture archiving and communication system monitor. The region of interest (ROI) marker in an oval shape is placed in the center of the mass (short arrows) to cover 50% of the largest tumor area. The mean value of precontrast CTHU is 53 and that of postcontrast CTHU is 103. Sinusitis is found in the left frontal sinus with mucus retention (black curved arrows); (b) Coronal postcontrast CT scan shows multiple masses or enlargement in various ophthalmic tissues with regular borders and homogeneous contrast enhancement in bilateral orbital cavities as follows: a mass at the extraconal compartment of the right orbital cavity (black arrow),the right lacrimal gland (black curved arrow), the left lacrimal gland (large white curve arrow), the right superior rectus muscle belly (small white curve arrow) and the right infraorbital nerve (white arrow). Sinusitis is noted in the right ethmoid sinus with mucus retention (arrowhead); (c) Pathologic specimen shows infiltration of many lymphoplasma cells and mild fibrosis (Hematoxylin-eosin stain, original magnification ×200); (d) Immunostaining for IgG expression shows many plasma cells are positive for IgG stains (original magnification x200); (e) Immunostaining for IgG4 expression shows abundant IgG4-positive plasma cells have infiltrated the lesion. IgG4-postive/IgG-positive plasma cell ratio is more than 40%. There are more than 100 IgG4-positive plasma cells in one high-powered field (>100/HPF) (original magnification x200).
Figure 2
(Please see attachment)
Figure 2. A 66-year-old man with orbital lymphoma shows a painless solitary lesion at the right lower eyelid. (a) Axial precontrast CT image shows a small nodule with an irregular border and homogeneous density involves the preseptal space of the right lower eyelid (arrowhead). Precontrast CT Hounsfield unit scale (CTHU) of the nodule is 57; (b) Axial postcontrast CT image shows the nodule demonstrates homogeneous enhancement (arrowhead). Postcontrast CTHU of the nodule is 68. Serum IgG4 level of the patient is 44.3 mg/dl.

Reviewer 2 Report
Authors present an original study on radiological and immunological criteria for Differentiation between benign IgG4-related orbital disease from malignant orbital lymphoma. Methods and results are fairly presented and several CT Features with elevated Serum IgG4 are found to be significant in distinguishing IgG4-ROD from OL. It would be interesting to add further noninvasive Methods in the Discussion part which could potentially help in further diagnostic workup (role of Orbit MRI). Furthermore it would be of interest in the Discussion and Conclusion part to add possible consequences of "prebiopsy" diagnosis - i.e. would the authors treat the patient who fulfills the criteria for IgG4-ROD with Steroid Trial without biopsy and if there are any prospective experiences of that Kind.
Author Response
Review Report Form
Open Review
( ) I would not like to sign my review report
(x) I would like to sign my review report
English language and style
( ) Extensive editing of English language and style required
( ) Moderate English changes required
(x) English language and style are fine/minor spell check required
( ) I don't feel qualified to judge about the English language and style
|
Yes |
Can be improved |
Must be improved |
Not applicable |
|
|
Does the introduction provide sufficient background and include all relevant references? |
(x) |
( ) |
( ) |
( ) |
|
Is the research design appropriate? |
(x) |
( ) |
( ) |
( ) |
|
Are the methods adequately described? |
(x) |
( ) |
( ) |
( ) |
|
Are the results clearly presented? |
( ) |
(x) |
( ) |
( ) |
|
Are the conclusions supported by the results? |
( ) |
(x) |
( ) |
( ) |
Comments and Suggestions for Authors
Authors present an original study on radiological and immunological criteria for Differentiation between benign IgG4-related orbital disease from malignant orbital lymphoma. Methods and results are fairly presented and several CT Features with elevated Serum IgG4 are found to be significant in distinguishing IgG4-ROD from OL. It would be interesting to add further noninvasive Methods in the Discussion part which could potentially help in further diagnostic workup (role of Orbit MRI).
Reply: Thank you for your comments. In addition to the previous descriptions of orbit MRI in Discussion, we added the more contents of orbit MRI in page 12-13, line 346-388 with marks in blue color (clean copy).
Page 12-13 line 346-388 (clean copy):
In addition to CT, magnetic resonance imaging (MRI) also helps in further diagnostic workup of orbital tumors and provides ocular anatomy for lesions involvement, perineural spread and intracranial extension [12]. Both retinoblastomas typically found in children and uveal melanomas in adults appear in the globe. Retinoblastoma is slightly hyperintense on T1 weighted MRI (T1WI) and very hypointense relative to vitreous on T2-weighted MRI (T2WI), and well contrast enhancement on postcontrast CT and contrast-enhanced (CE) MRI [12, 18]. Ninety percent of retinoblastomas demonstrate calcifications on precontrast CT scans [18]. Melanomas with melanin show characteristic hyperintensity on T1WI and hypointensity on T2WI [12, 18]. For intraconal orbital tumors, gliomas common among children result in fusiform enlargement of the optic nerve on axial CT and MRI [12, 18]. In contrast, meningiomas, commonly seen in the 5th decade of life, classically show the contrast-enhancing tumor with a “tram-tract” configuration alongside the optic nerve on axial postcontrast CT or CE MRI [12]. The most common benign orbital tumor in adults is a cavernous hemangioma, which typically demonstrates a well-defined dense unilateral orbital intraconal mass with intra-tumoral calcifications on precontrast CT scans and MRI. The enhancement spread pattern on a dynamic postcontrast CT and dynamic CE T1WI can help to distinguish between cavernous hemangioma and schwannoma [18, 19]. Cavernous hemangiomas show initial patchy enhancement on arterial phase but schwannomas start a wide area of enhancement. The most common congenital orbital nodules are dermoids, which usually show a well outlined round or oval tumor with a capsule, and low density or fat contents in the extraconal space on CT scans or MRI [18]. Due to fat contents, dermoids typically show hyperintensity on T1WI, hyperintensity on T2WI and hypointensity on short tau inversion recovery MRI (STIR) [18]. Benign mixed tumor of lacrimal gland usually seen in middle-aged patients demonstrates a well-circumscribed round or oval tumor with homogeneous enhancemenat on postcontrast CT and CE MRI [12]. Malignant epithelial lacrimal gland tumors show a mass with a well- or poor-defined margin with associated bony remodeling or destruction in 70% cases on CT scans [18].
Multiple or multi-compartmental orbital masses include venolymphatic malformations (VLM), rhabdomyosarcoma (RMS), plexiform neurofibroma, thyroid ophthalmopathy (TO), orbital pseudotumor (OP), lymphoma, metastases, and IgG4-ROD [18]. The first three types of masses are common among children; the last five, among adults [3, 5, 18]. VLM usually appears poorly-defined, lobulated and multiloculated lesions with various signal intensity on T1WI and T2WI [18]. VLM may demonstrate fluid-fluid level on MRI, which is highly suggestive of the diagnosis of VLM [12]. VLM, RMS and plexiform neurofibroma may have similar findings on CT and MRI [18]. TO causes enlarged bilateral myositis of the extraocular muscles, often involves medial and inferior rectus muscles with sparing tendinous insertions on CT and MRI [12], and is related to elevated thyroid-stimulating hormone level [18]. OP, IgG4-ROD and OL show similar MRI features on conventional sequences, which are hypointense on T1WI and T2WI and well contrast enhancement on postcontrast T1WI. Furthermore, diffusion-weighted imaging (DWI) with apparent diffusion coefficient (ADC) mapping can help to differentiate between benign and malignant orbital lesions [1, 18]. Sepahdari et al have reported that an ADC value < 1.0x 10-3 mm2/ sec and an ADC ratio < 1.2 are optimal for predicting orbital malignant tumors [1, 18]. Prior studies used ADC value < 1.0x 10-3 mm2/ sec and ADC ratio < 1.2 to differentiate orbital lymphoma from benign OP and IgG4-ROD with more than 95% accuracy [1, 18]. However, ADC values and ratios cannot differentiate OP from IgG4-ROD because the two disease have similar these values [1, 18].
Furthermore it would be of interest in the Discussion and Conclusion part to add possible consequences of "prebiopsy" diagnosis - i.e. would the authors treat the patient who fulfills the criteria for IgG4-ROD with Steroid Trial without biopsy and if there are any prospective experiences of that Kind.
Reply: Thank you for your advice. We added possible consequences of “prebiopsy” diagnosis and prospective experiences in Page 14, line 411-418 with marks in blue color (clean copy). Our conclusions have provided the special CT features and elevated serum IgG4 levels ≥ 132.5 mg/dl as testing tools for “prebiopsy” diagnosis of IgG4-ROD in page 14 line 430-441.
Page 14, line 411-418 as follows (clean copy):
In clinical practice, some IgG4-ROD patients may be at high risk for biopsy and/or refuse biopsy. However, once these patients meet possible diagnosis of IgG4-ROD [20], systemic steroid treatment may be a good alternative. The criteria for possible IgG4-ROD diagnosis include elevated serum IgG4 (≥135 mg/dl), enlargement of the lacrimal gland or masses, enlargement, or hypertrophic lesions in various orbital tissues [20]. Clinicians could forgo further biopsy if such patients respond well to glucocorticoids within weeks, such as reductions in the size of tumors, improvements of symptoms, and a significant decrease in serum IgG4 [20]. Alternative non-vital organ or lip biopsy may be an acceptable option [21].
Page 14 line 430-441 (clean copy):
Conclusions
This study compared prebiopsy precontrast and postcontrast CT features, serum IgG4 level and serum IgG level of IgG4-ROD with those of OL. The key findings showed that IgG4-ROD had high correlation with the presence of lesions with regular borders, presence of multiple lesions, involving the lacrimal gland, both lacrimal glands and bilateral orbital cavities on CT scans, higher values of postcontrast CTHU, postcontrast CTHU/precontrast CTHU ratios, serum IgG4 levels, and serum IgG4/IgG level ratios (all p < 0.05). For diagnosis of IgG4-ROD, postcontrast CTHU ≥ 89.5 and serum IgG4 level ≥ 132.5 mg/dl provided moderate diagnostic accuracy, AUC = 0.776, and 0.847, respectively, which were higher than those of postcontrast HU/precontrast HU and serum IgG4/IgG level ratio. The special CT features and elevated serum IgG4 levels could help differentiate IgG4-ROD from OL. Prebiopsy diagnosis of the uncommon solitary type of IgG4-ROD could also depend on a painless orbital mass and elevated CTHU ≥ 89.5 and serum IgG4 level ≥ 132.5 mg/dl.
